# Interaction of Oxicam Derivatives with the Artificial Models of Biological Membranes—Calorimetric and Fluorescence Spectroscopic Study

**DOI:** 10.3390/membranes12080791

**Published:** 2022-08-17

**Authors:** Jadwiga Maniewska, Żaneta Czyżnikowska, Berenika M. Szczęśniak-Sięga, Krystyna Michalak

**Affiliations:** 1Department of Medicinal Chemistry, Faculty of Pharmacy, Wroclaw Medical University, Borowska 211, 50-556 Wroclaw, Poland; 2Department of Inorganic Chemistry, Faculty of Pharmacy, Wroclaw Medical University, Borowska 211a, 50-556 Wroclaw, Poland; 3Department of Biophysics and Neuroscience, Faculty of Medicine, Wroclaw Medical University, T. Chałubińskiego 3a, 50-368 Wroclaw, Poland

**Keywords:** DSC, fluorescence spectroscopy, 1,2-benzothiazine derivatives, oxicams, piroxicam, model membranes, drug–membrane interaction, Laurdan, Prodan

## Abstract

The modified 1,2-benzothiazine analogues designed as new drug candidates and discussed in this paper are oxicam derivatives. Oxicams are a class of non-steroidal anti-inflammatory drugs (NSAIDs). Their biological target is cyclooxygenase (COX), a membrane protein associated with the phospholipid bilayer. In recent decades, it has been proven that the biological effect of NSAIDs may be closely related to their interaction at the level of the biological membrane. These processes are often complicated and the biological membranes themselves are very complex. Therefore, to study these mechanisms, simplified models of biological membranes are used. To characterize the interaction of six oxicam derivatives with DPPC, DMPC and EYPC, artificial models of biological membranes (multi-bilayers or liposomes), differential scanning calorimetry (DSC) and fluorescence spectroscopy techniques were applied. In spectroscopic measurements, two fluorescent probes (Laurdan and Prodan) localized in different membrane segments were used. All tested oxicam derivatives interacted with the lipid bilayers and may penetrate the artificial models of biological membranes. They intercalated into the lipid bilayers and were located in the vicinity of the polar/apolar membrane interface. Moreover, a good drug candidate should not only have high efficiency against a molecular target but also exhibit strictly defined ADMET parameters, therefore these activities of the studied compounds were also estimated.

## 1. Introduction

The interaction of drugs with biological membranes is a complex and pharmacologically extremely important process. This interaction is often during a preliminary stage in the body when drugs must cross biological membranes in order to be absorbed, and then undergo processes of distribution, metabolism and finally excretion. The interaction of different drugs, e.g., antibiotics and non-steroidal anti-inflammatory drugs (NSAIDs) with lipid membranes is never a trivial issue. Some antibiotics (e.g., daptomycin) and lipopeptides interact as a primary step with the lipid membrane. Their mechanism of action is strongly dependent on both the chemical structure of the peptide and on the lipid composition [1,2]. Overall, drugs can act on the surface of the cell membrane as well as have intracellular targets of action. The interaction of a drug with a biological membrane may affect the kinetics of the penetration of a molecule into the cell. Moreover, the pharmacological action of many drugs may be a consequence of their direct interaction with receptors which are specific membrane proteins. Drug molecules, by directly affecting the phospholipid environment of a membrane protein, may change both functions and biophysical properties of biological membranes, modifying an activity of an associated protein and thus influencing specific cellular processes [3].

NSAIDs are the most widely prescribed medications in the world [4]. The chemical advances of the 19th–20th centuries promoted the development of NSAIDs. Most of these drugs were initially organic acids (later on non-acidic compounds were discovered), which have since become the first-choice drugs for the treatment of various pain, fever and inflammation conditions [5]. Epidemiological, clinical and preclinical studies provide compelling evidence that those drugs also have antineoplastic properties [6,7,8] and may provoke a proapoptotic effect in cancer cells [9,10]. Moreover, the epidemiological studies suggested that long-term use of NSAIDs decreases the risk for Alzheimer’s disease as well as Parkinson’s disease by reducing neuroinflammation [11]. Understanding the interaction of NSAIDs with biological membranes is crucial for understanding both their pharmacokinetics and mechanism of action. The biological action of NSAIDs (therapeutic effect, but also a number of side effects) may result from their interaction with the cell membrane [12]. Drugs belonging to this group are mainly taken orally, and their biological target is cyclooxygenase (COX)—a membrane protein occurring in three isoforms (COX-1, COX-2 and COX-3). This enzyme is associated with the phospholipid bilayer surrounding the endoplasmic reticulum, as well as the cell nucleus [13].

The molecular mechanism of gastrointestinal damage caused by NSAIDs was considered by Lenard M. Lichtenberger et al. The studies were performed using model DPPC membranes and selected anti-inflammatory drugs: naproxen, indomethacin, diclofenac, aspirin and salicylic acid. The conducted studies proved that NSAIDs not only inhibit the formation of gastroprotective prostaglandins, but also create chemical bonds with neutral phospholipids that build the gastrointestinal mucosa. Then, the hydrophobic character of this specific barrier is weakened, and it is possible to damage the mucosa by hydrochloric acid present in the stomach, thus the occurrence of side effects from the gastrointestinal tract. The formation of a complex of NSAIDs with the zwitterionic phospholipid and its administration in this form may reduce the gastrotoxic effect of the drug by enhancing membrane penetration, as well as increasing the anti-inflammatory and antipyretic activity [14].

The idea that piroxicam, meloxicam and tenoxicam are highly sensitive to any changes in their environment (i.e., variable solvent parameters) was confirmed by the study of Rona Banerjee et al. [15]. The effect of piroxicam and meloxicam on model biological membranes using small unilamellar vesicles (SUVs) made of DMPG (dimyristoylphosphatidylglycerol) and DMPC (dimyristoylphosphatidylcholine) was also investigated by Hirak Chakraborty’s team. It turned out that the partition coefficient of piroxicam between the lipid phase and the water phase decreases with increasing DMPC content, while in the case of meloxicam the relationship is inverse [16]. The same team also carried out a study on the modulation of the properties of biological membranes by various ligands. The interaction of piroxicam with isolated mitochondria, membrane mimetic systems, intact cells and a mitochondrial protein cytochrome *c* was examined. It has been shown that piroxicam can modify the structure of the mitochondrial membrane, leading to its rupture and increasing its permeability. Moreover, studies on membrane mimetic systems such as DMPC liposomes and mixed micelles show that causing membrane fusion is a general property of piroxicam [17].

The thermal properties of four oxicam NSAIDs, meloxicam, tenoxicam, piroxicam and lornoxicam, were investigated using the spectroscopic and DSC methods combined with molecular modeling by Iwonne Kyrikou et al. The tests were performed on DPPC bilayers in a neutral and acidic (pH 2.5) environment. It has been shown that these drugs lower the temperature and widen the peak of main phase transition as well as abolish a pre-transition [18].

Binding of indomethacin, acemetacin and nimesulide to a model membrane made of DPPC was investigated by Marlene Lúcio et al. It was shown that each of the tested NSAIDs modifies the main phase transition depending on its location in the bilayer by influencing the main phase transition temperature, pre-transition temperature, heat capacity and enthalpy as well as the packing of membrane phospholipids. The derivatives of arylacetic acid (indomethacin, acemetacin), showed the highest destabilizing effects in the lipid gel phase and the greatest influence on the main phase transition temperature, most likely due to their better penetration into the bilayer. It was supposed that it may be a part of the mechanism by which these NSAIDs attenuate the hydrophobic barrier properties of the stomach’s mucus phospholipid gel bilayer with the consequent increase in the back diffusion of luminal acid into the mucosa and the development of erosions [19].

The interaction of ibuprofen, diclofenac and naproxen with lipid membranes was analyzed by Marcela Manrique Moreno’s team in a model membrane made of DMPC using a combination of various spectroscopic as well as thermodynamic methods. As it turned out, the interaction of DMPC with NSAIDs is mainly mediated by the phosphate groups of the hydrophilic “heads” of the membrane phospholipids [20].

The effect of nimesulide, tolmetin, piroxicam, meloxicam and indomethacin on model membranes made of DMPC was studied by the Claudia Nunes team. The X-ray scattering method was used to perform the experiments. The study confirmed the role played by the presence of high concentrations of NSAIDs in membranes in the development of gastrointestinal toxicity. This toxicity is exacerbated in an acidic environment. Among all the compounds tested, meloxicam (in appropriately low doses) showed the lowest gastrointestinal mucosa destabilizing effect, which may be related to its increased (practically selective) activity against cyclooxygenase 2 (COX-2). Meloxicam is also characterized by an exceptionally strong lipophilic character in the group of oxicams. Its selectivity towards COX-2 is, however, strictly dose-dependent, and damage to the gastrointestinal tract may occur after higher doses of this drug [21].

The interaction of piroxicam occurring in different tautomeric forms with phospholipid bilayers was investigated by Natalia Wilkosz et al. It was shown that all forms (neutral, zwitterionic and cationic) of piroxicam interact with the tested membranes. The effect of piroxicam on the liposomes in which the drug can be transported has also been investigated, and it has been proven that PEGylated liposomes can accommodate more of the drug through the ability of piroxicam to locate inside the lipid bilayer as well as on the PEG layer [22].

Catarina Pereira-Leite’s team aimed to describe the effects of diclofenac and naproxen on the permeability and structure of lipid bilayers. Their work supports the hypothesis that NSAID–lipid interactions, in particular at the mitochondrial level, may be another key step among the mechanisms underlying NSAID-induced cardiotoxicity [23].

There is a need for new, safer NSAIDs than currently those used, which will be used both as anti-inflammatory agents and also as potential chemopreventive agents. With long-term therapy, they could prevent the development of cancer, especially colorectal cancer in patients at increased risk of developing cancer [7,24].

Since oxicams are a group of NSAIDs with COX as their biological target of action found inside the cell and the drug molecules must penetrate cell membranes to reach it, it is important to investigate the interaction of these compounds with the phospholipid bilayer. The modified 1,2-benzothiazine derivatives discussed in this paper are oxicam derivatives, therefore studying their interaction with phospholipid membranes may help to understand their molecular mechanisms of action. Unfortunately, in biological systems, these processes are often complicated and the biological membranes themselves are very complex. Therefore, to study these mechanisms simplified models of biological membranes are used, e.g., phospholipid mono- or bilayers as well as liposomes [25]. In the present work, the results of calorimetric and fluorescence spectroscopy experiments are described. The influence of six analogues of piroxicam on the phase behavior of phospholipid multi-bilayers, and fluorescence quenching by these compounds of two fluorescent probes in liposomes—Laurdan and Prodan—were studied.

## 2. Materials and Methods

### 2.1. Chemicals

Phospholipids: 1,2-dipalmitoyl-*sn*-glycero-3-phosphatidylcholine (DPPC), 1,2-dimyristoyl-*sn*-glycero-3-phosphatidylcholine (DMPC) and egg yolk phoshphatidylcholine (EYPC) were obtained from Merck Life Science (Darmstadt, Germany) and used as delivered without any purification.

Piroxicam was purchased from Thermo Scientific (Ward Hill, MA, USA).

Studied oxicam derivatives (1,2-benzothiazine derivatives) were synthesized at the Department of Medicinal Chemistry, Wroclaw Medical University, by Berenika Szczęśniak-Sięga. Their purity was confirmed by ^1^H NMR, ^13^C NMR, FT-IR, HRMS and elemental analysis (C, H, N). The synthesis and analysis of these compounds were described previously elsewhere [26,27,28]. Chemical structures and symbols of studied compounds are shown in Table 1.

Fluorescent labels: 6-dodecanoyl-2-dimethylaminonaphthalene (Laurdan) and 6-propionyl-2-dimethylaminonaphthalene (Prodan) were purchased from Molecular Probes (Eugene, OR, USA). Fluorescent probes were dissolved in DMSO to obtain 1 mM stock solutions.

Since studied compounds were insoluble in water, their chloroform or DMSO solutions were used for experiments. All other chemicals used in this study were of analytical grade.

### 2.2. Experimental

#### 2.2.1. Differential Scanning Calorimetry (DSC)

For each sample, 2 mg of phospholipid was dissolved in an appropriate amount of studied compound stock solution (5 mmol/L, in CHCl_3_) to obtain the desired compound: lipid molar ratio. The oxicam: phospholipid mixtures were studied at 0.05, 0.10, 0.15 and 0.20 molar ratios because in these concentrations in the former experiments with other oxicam derivatives (on the same calorimeters) we observed pronounced changes in thermal behavior of phospholipids [29]. All samples were dried under the stream of nitrogen and placed under vacuum for at least 1.5 h to evaporate the chloroform. Then, 15 µL of 20 mmol/L Tris-HCl buffer (20 mM Tris, 0.5 mM EDTA, 150 mM NaCl, pH 7.4) was added to each sample. Hydrated mixtures were heated to the temperature ca. 10 °C higher than main phase transition temperature of a given lipid and vortexed until a homogeneous dispersion was obtained. Samples were sealed in aluminum pans and scanned at the rate of 1.25 °C/min using a *Rigaku* calorimeter (Rigaku, Tokyo, Japan) in DMPC measurements or at the rate of 1 °C/min using *Unipan* microcalorimeter type 600 (Warsaw, Poland) in DPPC measurements. For each oxicam: phospholipid molar ratio at least 2 samples were prepared, and each sample was scanned at least 4 times. Calorimetric data were collected and processed offline using software developed by M.Sc. Łukasz Fajfrowski.

#### 2.2.2. Fluorescence Spectroscopy

Unilamellar EYPC liposomes were obtained by sonification of 2 mM/L phospholipid suspension in the same buffer solution as used in DSC experiments (pH 7.4) using a UP 200 s sonicator (*Dr. Hilscher*, GmbH, Berlin, Germany).

Fluorescent dyes: Laurdan and Prodan stock solutions (1 mM) were prepared in DMSO. The stock solutions of the studied compounds (30 mM) were also prepared in DMSO. The dispersion of EYPC liposomes was incubated with the fluorescent dye in darkness for 30 min at room temperature then the studied compound was added and liposomes were incubated for another 20 min (also in darkness, at room temperature). In all of the experiments the final EYPC concentration was 200 μM. The concentration of the fluorescent dye (Laurdan or Prodan) was 5 μM. Concentrations of studied compounds in the samples were within the range of 25–125 μM. The fluorescence experiments were performed with an LS 50B spectrofluorometer (*Perkin-Elmer* Ltd., Beaconsfield, UK) equipped with a xenon lamp using emission and excitation slits of 5 nm. The excitation wavelength for Laurdan was 390 nm and for Prodan was 360 nm. The recorded fluorescence spectra were processed with FLDM Perkin-Elmer 2000 software. The studied compounds alone did not exhibit fluorescence in the spectral region of interest.

Both calorimetric and spectroscopic measurements were performed in laboratories of the Department of Biophysics, Wroclaw Medical University (at present Department of Biophysics and Neuroscience).

#### 2.2.3. Prediction of ADMET Properties

The physicochemical properties, pharmacokinetics and ADMET activity of designed 1,2-benzothiazine derivatives were estimated based on the comprehensive database ADMETlab (2.0) [30].

## 3. Results

### 3.1. Differential Scanning Calorimetry

Differential scanning calorimetry (DSC) is a relatively fast and easy research method showing how the studied compounds modify the phase transition profile of phospholipids [31]. In our calorimetric experiments, the artificial model of biological membranes used in the investigation of the influence of the studied compounds (Table 1) on the thermal properties of two phospholipids (DPPC or DMPC) was the multi-bilayers one, obtained by the hydration of dry phospholipid film with simultaneous heating and vortexing the suspension (see Section 2.2.1).

#### 3.1.1. DSC Measurements of DPPC with Studied Compounds

The impact of the studied compounds on the lipid thermal behavior is presented in Figure 1, showing exemplary thermograms of DPPC mixed with compounds PR1, PR2, PR38 and PR12 at different molar ratios.

The addition of the oxicam derivatives caused vanishing of the DPPC pre-transition peak and a concentration-dependent shift of the main transition temperature towards lower values, which was accompanied by a decrease in transition peak area and broadening of the peak.

Moreover, at the PR2: DPPC and PR12: DPPC different molar ratios (see Figure 1b,d), the main peak appears to be composed of two overlapping peaks. This indicates that the main phase transition of the phospholipid is less cooperative when the lipid is mixed with compound PR2 or PR12.

The dependencies of the main transition temperature (Tm), the transition peak width at half-height (∆T_1/2_) and the transition enthalpy (ΔH) on the oxicam derivative: lipid molar ratio obtained for the mixtures of DPPC with oxicam derivatives are shown in Figure 2a–c, respectively. All examined compounds decreased the main transition temperature (Tm) of DPPC in a concentration-dependent manner (see Figure 2a).

The studied compounds exerted a markedly greater effect on the thermotropic properties of the DPPC multi-lamellar structures than piroxicam. The addition of the studied compounds to phospholipids also caused broadening of the transition peaks, which was visible as an increase in the transition peak half-width (Figure 2b). This process is accompanied by shifting the phase transition temperature (Tm) towards lower values, and this shift is also dependent on the concentration of the studied compounds. Phase transition profiles were most effectively broadened by PR12.

#### 3.1.2. DSC Measurements of DMPC with Studied Compounds

The exemplary thermograms of DMPC mixed with PR38, PR12, PR26 and PR27 at different molar ratios are shown in Figure 3. The addition of the oxicam derivatives also caused the vanishing of the phospholipid pre-transition peak and a concentration-dependent shift of the main transition temperature towards lower values, accompanied by a decrease in the transition peaks’ area and the broadening of the peaks. These effects were much more pronounced for DMPC than for DPPC model membranes.

In the case of the 0.15 and 0.20 molar ratios of PR27: DMPC (see Figure 3b,d), the main peak was so low that it was not possible to determine all thermotropic parameters of the main phase transition (see Figure 4, orange curve).

#### 3.1.3. Comparison of the Results Obtained in DSC Studies for DPPC and DMPC

All oxicam derivatives studied perturbed the DPPC and DMPC multi-bilayer structure. The effect was more pronounced for DMPC model membranes than for DPPC ones (see Figure 5). It might be explained by the fact that DMPC as a phospholipid with shorter acyl chains than DPPC is characterized by weaker interactions between hydrocarbon chains and more loosely-packed structures, which could provoke easier oxicam derivative incorporation into bilayers formed from DMPC.

The extent of changes induced by oxicam derivatives in parameters characterizing lipid phase transition was greater for DMPC phosphatidylcholine possessing shorter acyl chains than for DPPC with longer chains. In both lipids, the transition enthalpy was decreased by PR1, PR12 and PR27 more effectively than by PR2, PR26, PR38 and piroxicam, but the effect was more pronounced for the DMPC (Figure 5c). In both lipids, the main transition temperature was decreased by PR2, PR12 and PR27 more effectively than by PR1, PR38, PR26 and piroxicam, but the effect was also more pronounced for the DMPC (Figure 5a). T_m_ was similarly lowered by all studied compounds. The PR2 and PR27 in the DMPC bilayers were the most effective. The only exception was PR38, which induced a larger effect on DPPC. The presence of oxicam derivatives in the phosphatidylcholine model membranes also caused a broadening of transition peaks that was most pronounced for PR12 and PR27 in the DMPC bilayers (Figure 5b).

### 3.2. Fluorescence Spectroscopy

In our spectroscopic experiments, the liposomes obtained from egg yolk phosphatidylcholine (EYPC) were used as an artificial model of biological membranes. In these measurements, two fluorescent probes (Laurdan and Prodan) localized in different membrane segments were applied. According to Joseph Lakowicz, if the molecular location of a fluorescent probe within the lipid bilayer is known, quenching studies can be used to reveal the location of quenchers in the membrane [32]. Laurdan and Prodan both possess the same fluorophore connected to an alkyl chain of different lengths (three carbon atoms in the Prodan molecule and twelve in Laurdan). Therefore, Prodan molecules locate closer to the hydrophilic surface of a bilayer [33,34] than Laurdan, in which the fluorophore is located closer to the phospholipid glycerol groups [35], (see Figure 6).

The addition of all studied compounds to EYPC liposomes incubated with fluorescent probes resulted in quenching both Laurdan and Prodan fluorescence. For both fluorescent probes, the Stern–Volmer plots representing fluorescence quenching were linear in the presence of all studied compounds (Figure 7, Figure 8 and Figure 9).

Stern–Volmer plots for piroxicam (PRX)-induced quenching of Laurdan and Prodan fluorescence are also provided for comparison. Bars represent standard deviations of three independent experiments. Where no error bars are shown, it means that they were smaller than the symbols representing the results.

Stern–Volmer plots for piroxicam-induced quenching of Laurdan and Prodan fluorescence are also provided for comparison. Bars represent standard deviations of three independent experiments. Where no error bars are shown, it means that they were smaller than the symbols representing the results.

The results of fluorescence spectroscopy experiments were grouped according to the similarities in chemical structure of the tested compounds (PR1 and PR2—Figure 7, PR12 and PR38—Figure 8, PR26 and PR27—Figure 9).

The compounds caused strong quenching of fluorescence of both probes, although the effect was much more pronounced in the case of Laurdan for PR2, PR12 and PR27 and in case of Prodan for piroxicam, PR1, PR26 and PR38.

The experiments presented above revealed that the two-carbon aliphatic linker with a carbonyl group between 4-phenylpiperazine and 1,2-benzothiazine (present in compounds PR2, PR12 and PR27) increased the ability of the compounds to locate deeper in model membranes (where Laurdan is located). This may suggest an interaction between the lone pair of electrons of the carbonyl group and the model membrane, while the three-carbon linker between 4-phenylpiperazine and 1,2-benzothiazine caused the interaction with the regions closer to the surface of the phospholipid bilayer (where Prodan is located). Comparing the chemical structure of PR26 with PR27, and PR12 with PR38, which have the fluorine substituent in the same position but differ in type of the linker, it is clear that the type of the linker between 4-phenylpiperazine and 1,2-benzothiazine determines the location of the compounds in the lipid bilayer. Additionally, the position of fluorine substituent in the molecule determines the strength of interaction with the phospholipid bilayer.

### 3.3. Prediction of ADMET Properties

It is well known that a good drug candidate should not only have high efficiency against a molecular target but also exhibit strictly defined ADMET (adsorption, distribution, metabolism, excretion and toxicity) parameters. It is also proven that simple physicochemical properties such as molecular weight (MW), the number of hydrogen bond donors (nHD) and acceptors (nHA), hydrophobicity and polarity of compounds influence their in vivo behavior with particular consideration of solubility, metabolic stability and permeability. Therefore, in the present paper the physicochemical properties, pharmacokinetics and ADMET activity of designed 1,2-benzothiazine derivatives were estimated based on the comprehensive database ADMETlab (2.0) [30]. Results are presented in Table 2 and Table 3 and in Appendix A.

The molecular weight of drugs can impact various processes such as absorption, brain barrier penetration (optimal MW 80 Da) or elimination from the body. Additionally, an analysis of properties of marketed oral drugs showed that a particularly high number of hydrogen bond donors (nHD) may cause poor bioavailability and membrane permeability of small molecules [36]. Another important factor is topological surface area (TPSA). Many predictive models showed that an increase in TPSA decreases membrane diffusion, especially in the case of blood–brain barrier penetration [37]. In addition, several ADMET parameters are determined by using the lipophilicity of compounds, expressed as logP and logD. Data show that too-low lipophilicity decreased membrane permeability. On the other hand, problems with drug metabolism arise when lipophilicity is high. Due to the penetration of biological membranes, highly lipophilic compounds may be trapped in the bilayer [38].

In our study, the molecular weight of all designed compounds does not exceed 522 KDa. The number of hydrogen bond acceptors is less than eight and hydrogen bond donors is zero, as presented in Table 2. The TPSA of the designed compounds in most cases is less than or equal to 95. As can be seen, compounds PR2, PR12 and PR27 exhibit optimal values of logP at physiological pH. In addition, most of the compounds, except PR1 and PR2, exhibit good bioavailability and low blood–brain barrier permeability (BBB). Unfortunately, all 1,2-benzotiazine derivatives are characterized by a high plasma protein binding index—above 90%—which may affects their low therapeutic index. The analysis of properties of the designed compounds also showed that all of them may exhibit high passive MDCK (Madin−Darby canine kidney cells) permeability and low Caco-2 (colon adenocarcinoma cell lines) permeability (see Appendix A). MDCK and Caco-2 cell permeabilities are often used in laboratories to analyze transporter-mediated mechanisms of permeability. As the results presented in Appendix A show, the compounds PR1 and PR38 are potential inhibitors of CYP2D6. No investigated compounds showed inhibitory activity towards CP1A2. The analysis of toxicity is presented in Appendix A. According to the results of our in silico studies, all compounds exhibit high probability to be hepatotoxic. However, there is low probability of mutagenicity (AMES toxicity) and carcinogenicity of these compounds.

## 4. Discussion

The interaction of drug molecules with biological membranes can be an important factor influencing the pharmacokinetic parameters in therapy. The activity of protein receptors may also be modulated by lipids in their microenvironment. Moreover, in some pathological conditions changes in protein and lipid composition of membrane can be observed. Sometimes, drug molecules must cross the cell membrane barrier to reach their biological target, e.g., NSAIDs which bind to the cyclooxygenase, the intracellular enzyme [39].

NSAIDs may disturb and change the physicochemical properties of the biological membrane with which they interact. These interactions can appear as changes in membrane fluidity or permeability and be a result of specific interactions between a drug and phosphatidylcholine, one of the major membrane-building phospholipids. The described phenomenon is closely dependent on pH and is especially marked in acidic pH, present, for example, in membranes of gastric cells. This may be related to the gastrotoxic effect of most NSAIDs, which is partly due to their mechanism of action (inhibition of the synthesis of prostaglandins that protect the gastrointestinal tract) and partly due to the physicochemical properties of specific NSAIDs [40].

The effectiveness as well as the safety of NSAIDs are also determined by differences in their physicochemical properties, such as: water solubility, partition coefficient (logP) and dissociation constants (pKa). Most NSAIDs are weak acids (their dissociation constants range from 3 to 5) [18].

In recent decades, it has been proven that the biological effect of NSAIDs can be closely related to their interaction at the level of the biological membrane. This resulted in a dynamic development of in vitro research on these interactions. In this type of study, the artificial models of biological membranes, cell cultures and other techniques and molecular stimulations are commonly used [12].

To optimize and modulate the biological effects of NSAIDs, some efforts are made to synthesize the new derivatives of these drugs. In the present work we describe the results of calorimetric and fluorescence spectroscopic studies on six analogues of piroxicam from the group of oxicams (1,2-benzothiazine derivatives), named PR1, PR2, PR12, PR26, PR27 and PR38. The effect of these compounds on properties of the phase transition of phospholipid membranes and quenching of fluorescence of two fluorescent probes, Laurdan and Prodan, were studied. It is important that the location of the applied fluorescent probes within membranes is well known.

It was previously shown in our studies that oxicam derivatives intercalate into the lipid bilayers and are located in the vicinity of the polar/apolar membrane interface. The details of drug–lipid interactions depend, however, both on the substituents present at the 1,2-bezothiazine scaffold as well as on the type of phospholipid used to prepare the model membrane [18,29,41,42,43].

On the basis of performed calorimetric measurements, the influence of tested oxicam derivatives on phospholipid bilayers formed of DPPC or DMPC and their influence on thermotropic properties of tested phospholipids can be listed in the following order: PR27 > PR12 > PR2 > PR26 > PR1 > PR38 > PRX. All studied new 1,2-benzothiazine derivatives influence the parameters of the main phase transition of phospholipids, building model membranes in greater extent than piroxicam. These results indicated a significant role of the phenylpiperazine substituent (not present in the piroxicam molecule) in enhancing of this interaction. Moreover, the two-carbon aliphatic linker with a carbonyl group between phenylpiperazine and 1,2-benzothiazine seems to also be quite an important structural element of the tested compounds, determining their influence on the phospholipid membranes. In addition, the presence of fluorine substituent in the benzene ring of the side chains at the *ortho* (PR12) or *para* position (PR27) seems to enhance an interaction with the studied lipid membranes. However, this effect disappears when there is no two-carbon aliphatic linker with a carbonyl group between phenylpiperazine and 1,2-benzothiazine in the molecule, such as in compounds PR26 and PR38.

Spectrofluorimetric measurements confirmed that oxicam derivatives affect the studied model membranes. The spectral properties of both fluorescent probes, Laurdan and Prodan, used in the experiments depend strongly on the amount of water penetrating appropriate regions of the bilayer under consideration (hydrophobic–hydrophilic interface in case of Laurdan and polar headgroup region in case of Prodan). These spectral properties depend also on the dynamics of water molecules, especially on the solvent dipolar relaxation process occurring in the vicinity of a fluorescent label [33]. Phospholipid EYPC used in the spectroscopic experiments was in a liquid-crystalline state during measurements carried out at room temperature.

The decrease in fluorescence intensity of the probes in the presence of oxicam derivatives may be a result of an interaction of the studied compounds with the used fluorescent probes. This interaction may affect the molecular organization of the phospholipid bilayer, and in that way influences the fluorophores’ microenvironment. The more pronounced quenching of the Prodan fluorescence than the Laurdan one exhibited by piroxicam, PR1, PR38 and PR26 indicates that the bilayer region where Prodan is located is more affected by these compounds. On the contrary, compounds PR2, PR12 and PR27 quenched the fluorescence of Laurdan more strongly than of the Prodan one. This is most likely caused by the difference in the chemical structure of the studied compounds, which determined their location in a phospholipid bilayer. The experiments carried out by fluorescence spectroscopy have shown that the presence of a two-carbon aliphatic linker with a carbonyl group in PR2, PR12 and PR27 structures as well as fluorine substituent in compounds PR12 and PR27 seems to increase the ability of these compounds to quench Laurdan fluorescence in the liquid-crystalline state of a phospholipid membrane.

The prediction of drug-likeness and pharmacokinetic in silico properties of the oxicam derivatives showed that all compounds exhibit slightly different properties than piroxicam (see https://go.drugbank.com/drugs/DB00554 (accessed on 15 June 2022)). All of them have a higher molecular weight, which can affect their ability to permeate the intestinal epithelium through the pores between the cells. On the other hand, they are characterized by a lower TPSA value as compared to piroxicam (108 Å^2^), which allows to predict their high capacity for intestinal absorption. All compounds met Lipinski’s rule of five and followed the criteria for orally active drugs. The estimated values of logP for all considered 1,2-benzothiazine derivatives were higher than in the case of piroxicam (logP = 3.06). More hydrophobic molecules will be preferentially localized in cells in their hydrophobic compartments, such as the hydrophobic part of lipid bilayers. As a result, these properties determine the bioavailability of the studied compounds.

## 5. Conclusions

In the present work, calorimetry and fluorescence spectroscopy were used to study the interactions of oxicam analogues with lipid bilayers. The obtained results indicated that the studied compounds interact with lipid bilayers and may penetrate lipid membranes. In lipid bilayers, they are located in the vicinity of the polar/apolar membrane interface. The fluorescence spectroscopy experiments revealed that compounds PR1, PR38 and PR26 locate close to the hydrophilic surface of a bilayer, while PR2, PR12 and PR27 locate closer to the glycerol groups of the phospholipid. We presume that deeper penetration into the phospholipid bilayer and stronger influence on the model membranes may be related to the presence of a two-carbon aliphatic linker with a carbonyl group between 1,2-benzothiazine and phenylpiperazine in the molecule (PR2, PR12 and PR27). Additionally, this effect may be enhanced by the fluorine substituent in the benzene ring of the side chain of the molecule (PR12 and PR27). These interactions could be responsible for at least some of the membrane-dependent effects of oxicam analogues; however, the understanding of their importance needs further studies (e.g., the evaluation of the effect of these derivatives on phase-separated lipid systems).

## Figures and Tables

**Figure 1 membranes-12-00791-f001:**
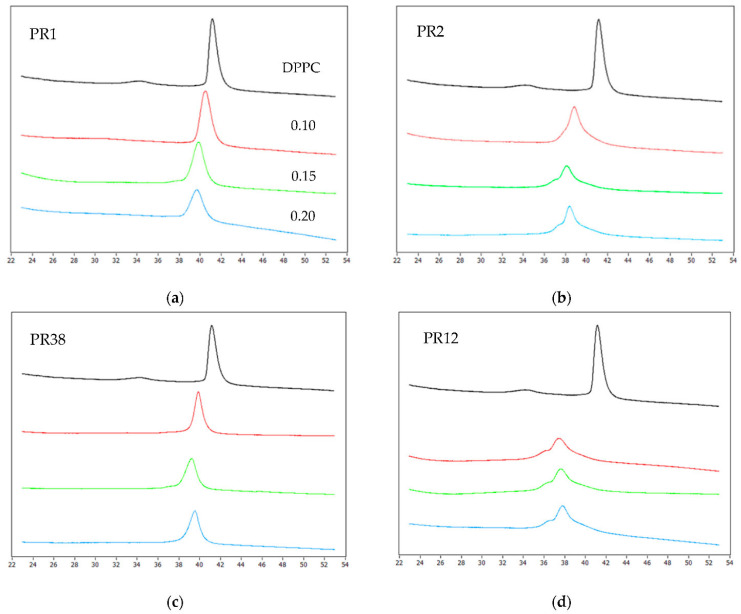
The exemplary thermograms obtained for DPPC mixed with compounds: (**a**) PR1, (**b**) PR2, (**c**) PR38, (**d**) PR12 as well as for pure lipid (the first curve from the top—black color). Curves in the figure represent the thermograms obtained for different compounds. DPPC molar ratios from the top to the bottom: 0 (pure lipid), 0.1, 0.15, 0.2.

**Figure 2 membranes-12-00791-f002:**
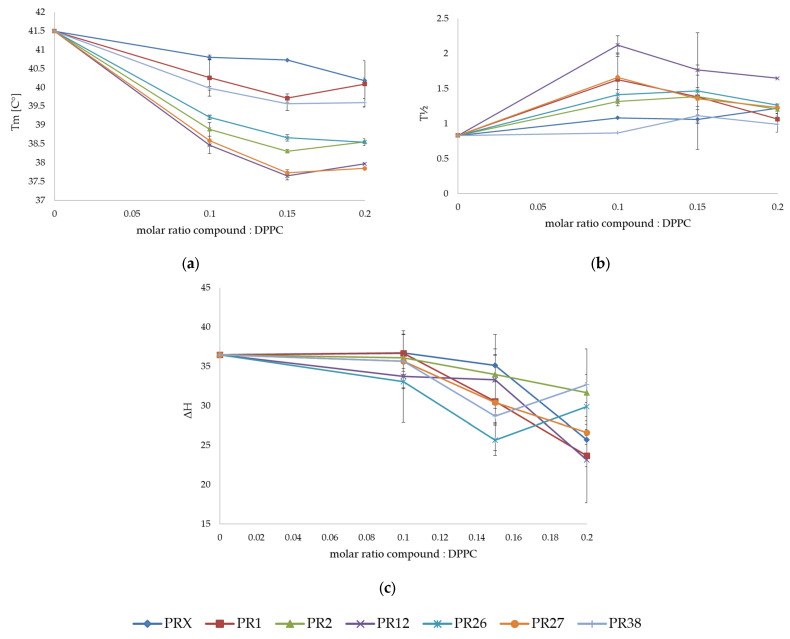
Influence of studied compounds on the parameters of DPPC main phase transition: temperature Tm (**a**), peak half-width ∆T_1/2_ (**b**) and phase transition enthalpy ∆H (**c**); mean ± SD, *n* = 8.

**Figure 3 membranes-12-00791-f003:**
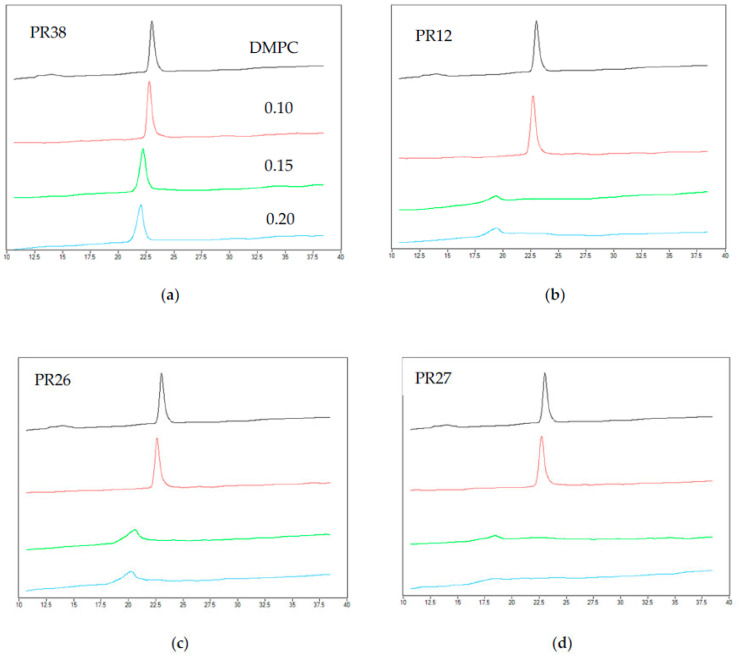
The example of thermograms obtained for DMPC mixed with compounds: (**a**) PR38, (**b**) PR12, (**c**) PR26, (**d**) PR27 as well as for pure lipid (the first curve from the top—black color). Curves in the figure represent the thermograms obtained for different molar ratios of the studied compound: DPPC (from the top to the bottom 0, 0.1, 0.15, 0.2).

**Figure 4 membranes-12-00791-f004:**
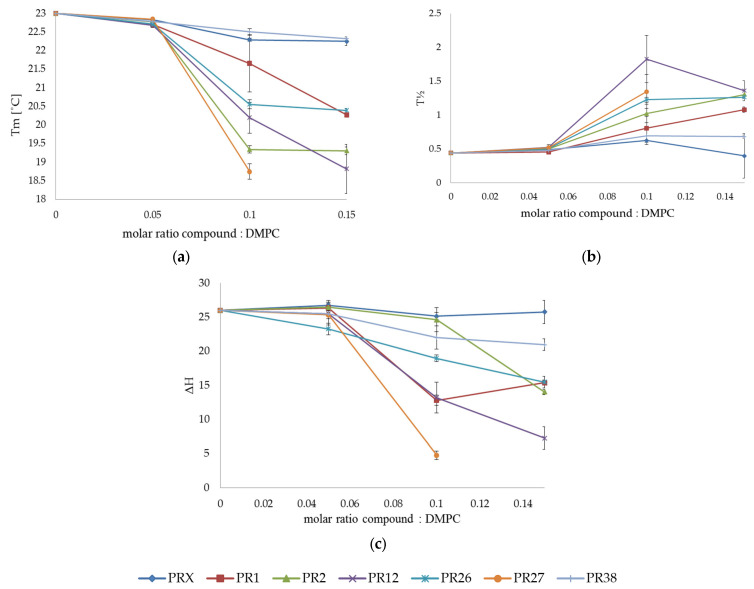
Influence of studied compounds on the parameters of DMPC main phase transition: temperature (**a**), half-width of the peak (**b**) and enthalpy of phase transition (**c**); mean ± SD, *n* = 8.

**Figure 5 membranes-12-00791-f005:**
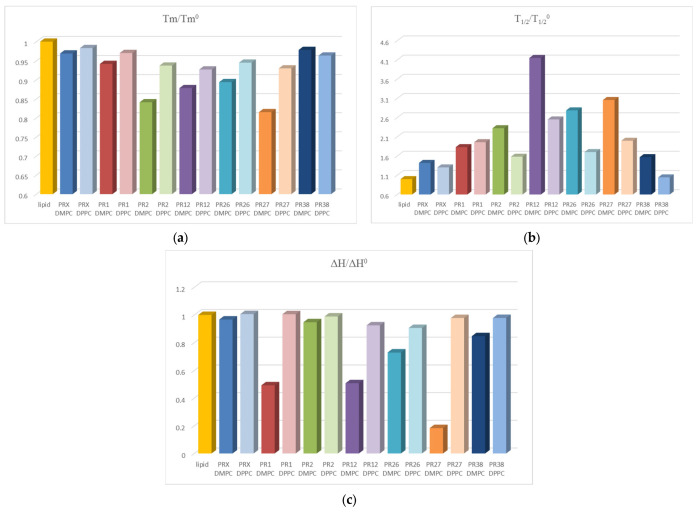
Relative parameters of phospholipid main phase transition: main transition temperature Tm (**a**), transition peak width at half-height T_1/2_ (**b**), transition enthalpy ∆H (**c**) for DPPC and DMPC in the presence of studied compounds at compound:phospholipid molar ratio 1:10.

**Figure 6 membranes-12-00791-f006:**
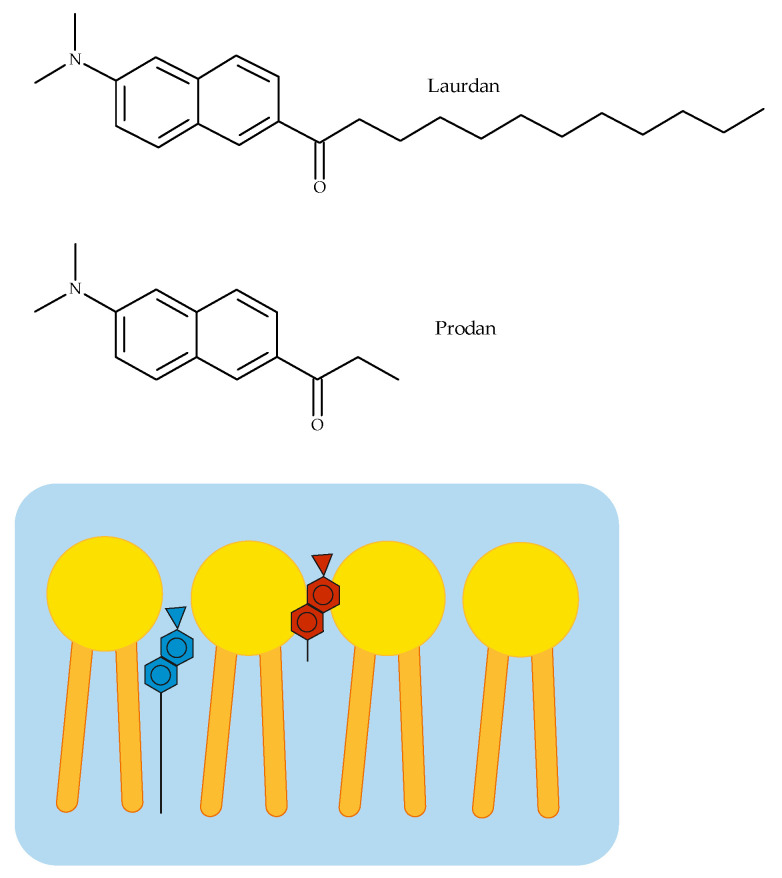
Chemical structure and location of Laurdan and Prodan in the phospholipid bilayer. Prodan molecules (red) locate closer to the hydrophilic surface of a bilayer than Laurdan (blue), in which the fluorophore is positioned closer to the phospholipid glycerol groups [33].

**Figure 7 membranes-12-00791-f007:**
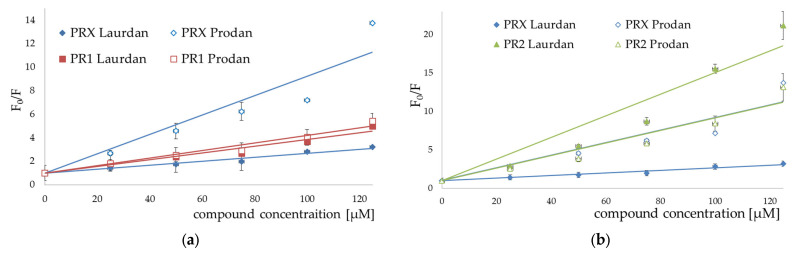
Stern–Volmer plots for quenching of Laurdan (full symbols) and Prodan (open symbols) fluorescence induced by PR1 (**a**) and PR2 (**b**); mean ± SD, *n* = 3.

**Figure 8 membranes-12-00791-f008:**
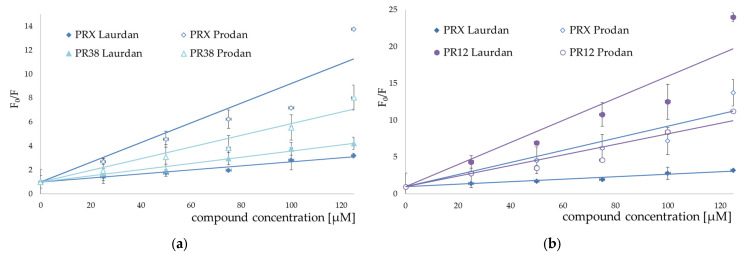
Stern–Volmer plots for PR38 (**a**) and PR12 (**b**) induced quenching of Laurdan (full symbols) and Prodan (open symbols) fluorescence; mean ± SD, *n* = 3.

**Figure 9 membranes-12-00791-f009:**
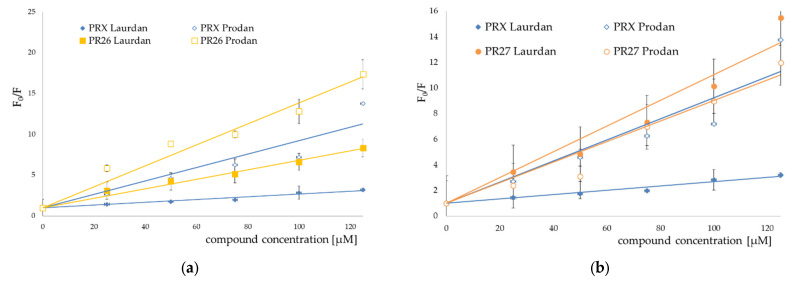
Stern–Volmer plots for PR26 (**a**) and PR27 (**b**) induced quenching of Laurdan (full symbols) and Prodan (open symbols) fluorescence; mean ± SD, *n* = 3.

**Table 1 membranes-12-00791-t001:** Chemical structures and symbols of studied compounds.

Compound Symbol	Chemical Structure
PR1	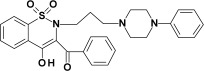
PR2	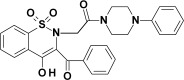
PR12	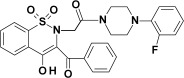
PR26	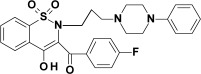
PR27	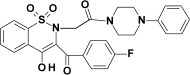
PR38	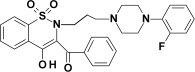
PRX (piroxicam)	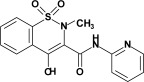

**Table 2 membranes-12-00791-t002:** Physicochemical properties of studied compounds.

Parameter/ Optimal Value	Compound					
	PR1	PR2	PR12	PR26	PR27	PR38
MW (molecular weight) optimal 100–600	503.19	503.15	521.14	521.18	521.14	521.18
nHA (number of hydrogen bond acceptors) optimal 0–12	7	8	8	7	8	7
nHD (number of hydrogen bond donors) optimal 0–7	0	0	0	0	0	0
TPSA (topological polar surface area) optimal 0–140	78	95	95	78	95	78
nRot (number of rotatable bonds) optimal 0–11	7	6	6	7	6	7
nRing (number of rings) optimal 0–6	5	5	5	5	5	5
nHet (number of heteroatoms) optimal 1–15	8	9	10	9	10	9
logP (log of the octanol/water partition coefficient) optimal 0–3	3.9	3.1	3.3	4.0	3.3	4.0
logD (logP at physiological pH) optimal 1–3	3.3	2.4	2.5	3.3	2.4	3.2

**Table 3 membranes-12-00791-t003:** Medicinal chemistry of studied compounds.

**Parameter**/ **Optimal Value**	**Compound**					
	**PR1**	**PR2**	**PR12**	**PR26**	**PR27**	**PR38**
QED (measure of drug-likeness based on the concept of desirability; attractive > 0.67, unattractive 0.49–0.67, too complex < 0.34)	0.4	0.4	0.4	0.3	0.4	0.3
SA score (synthetic accessibility score is designed to estimate ease of synthesis of drug-like molecules; ≥6—difficult, <6—easy to synthesize)	3.0	3.0	3.0	3.1	3.0	3.1
Fsp3 (number of sp3 hybridized carbons/total carbon count, correlating with melting point and solubility; ≥0.42 is considered a suitable value)	0.3	0.2	0.2	0.3	0.2	0.2
Lipinski Rule (MW ≤ 500; logP ≤ 5; Hacc ≤ 10; Hdon ≤ 5; if two properties are out of range, a poor absorption or permeability is possible, one is acceptable)	all accepted					
Pfizer Rule (compounds with a high log *p* (>3) and low TPSA (<75) are likely to be toxic)	all accepted					
GSK Rule (MW ≤ 400; logP ≤ 4; compounds satisfying the GSK rule may have a more favorable ADMET profile)	all rejected					

## Data Availability

Not applicable.

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
