# Peer review of "Interaction of Oxicam Derivatives with the Artificial Models of Biological Membranes—Calorimetric and Fluorescence Spectroscopic Study"

_membranes, 2022, doi:10.3390/membranes12080791_

Round 1
Reviewer 1 Report
The research article by K. Michalak et al describes a detailed study about the effects induced by oxicam derivatives on model lipid membrane systems through calorimetric and fluorescent spectroscopy analysis. The experimental evidence confirm the oxicam derivatives interact with both lipid systems and show a different partitioning within the lipid bilayer - close to the hydrophilic surface of the bilayer or close to the glycerol group of the phospholipid. The different penetration behavior of the molecules has been correlated to the presence of a specific aliphatic linker including a carbonyl functional group and the effect can be enhanced by the presence of fluorine substituent.
In general the manuscript is well-written, organized and clear. The experimental procedures are systematic and supported by the statistics. Figures are generally clear and appealing even if the graphics of some of them can be improved. However, some points need to be addressed and clarified before considering, from my side, the manuscript suitable for Membranes journal. Below, the list of points need to be addressed
-The title should be reviewed; the study is mainly experimental so I suggest to remove 'In Silico study' from the title.
-In the introduction authors mention the drug-membrane interaction as a crucial and complex process which may regulate the drug activity; then a detailed and exhaustive description of NSAIDs has been provided. I do think that the importance of the drug-membrane interaction should be even more highlighted by referring to antimicrobial and lipo-peptides (AMPs and LPs), known to interact as a primary step with the lipid membrane. Moreover, their mechanism of action is strongly dependent on both the chemical structure of the peptide and on the lipid composition; these considerations are coherent with the topic of the article and would broaden the discussion beyond the NSAIDs and thus the audience.
-Following the previous point, you should make few consideration on lipopeptide Daptomycin both for its mechanism of action and for its chemical structure which includes Kynurenine an highly fluorescent residue used to localize the peptide-membrane interaction (see following works https://doi.org/10.1021/acs.jpcb.0c06640; https://doi.org/10.1007/s00249-020-01445-w). You should cite this and similar references; tracing the position of drugs within lipid membranes is never a trivial issue and, in this way you would emphasize your work by referring to similar effects with molecules different than NSAIDs.
-Figure 1 graphics should be improved; the axis size is quite small and labels indicating the kind of derivatives as well as the molar ratio is lacking (the reading is quite tough with such info only in the caption)
-Same consideration are valid for Figure 3.
-In 3.1.3 lines 280-81 authors state: "In both lipids, the transition enthalpy was decreased by PR1, PR12 and PR27 more effectively than by PR2, PR26, PR38 and piroxicam (Figure 5c). " This is not coherent with Figure 5c, which show a different trend depending on the lipids: the decrease seems to be more effective on DMPC. Please clarify
- In 3.1.3 lines 281-83 authors state: "Also in both lipids,the main transition temperature was decreased by PR2, PR12 and PR27 more effectively than by PR1, PR38, PR26 and piroxicam (Figure 5a). " Also here the message is not totally coherent with the figure; panel 5a indicates a higher transition temperature decrease with the mentioned derivatives but only on DMPC. Please clarify
- In 3.1.3 lines 283-85 authors state: "Tm was reduced similarly by all studied compounds. The only exceptions were PR2 and PR27, which were the most effective in DMPC bilayers." Again, from the figure it seems that the larger effect is induced by PR2 and PR27 on DMPC lipids but this is not an exception; rather, PR38 seems as an exception as its effect is larger on DPPC. Please clarify
-lines 333-35 authors state: "The experiments presented revealed that the two–carbon aliphatic linker with a carbonyl group between 4-phenylpiperazine and 1,2-benzothiazine (present in compounds PR2, PR12 and PR27) increased the ability of the compounds to interact deeper" You may add a speculative sentence about the possible role of the charge due to the presence of carbonyl group.
-line 369: where does 586KDa come from? Is it MW of PRX - is not present in the tabel. Please clarify
-line 371 author state: "As can be seen compounds PR2 and PR12 exhibit optimal values of logP " Also PR27 show optimal value.
-As conclusive remarks, authors should refer as well to the possibility to evaluate effect of such derivatives forms also on phase-separated lipid systems. This can help in further understanding of the mechanism of action; even a brief referring to such an aspect would highlight the significance of the issue
Minor
-lines 61-62, use the abbreviation NSAIDs;
-line 235, may you refer to Figure 1b and 1d instead of Figure2;
-line 421; depend instead of dependent.
-line 471: "It can in result determine their bioavailability". Please, rephrase.
Author Response
The research article by K. Michalak et al describes a detailed study about the effects induced by oxicam derivatives on model lipid membrane systems through calorimetric and fluorescent spectroscopy analysis. The experimental evidence confirm the oxicam derivatives interact with both lipid systems and show a different partitioning within the lipid bilayer - close to the hydrophilic surface of the bilayer or close to the glycerol group of the phospholipid. The different penetration behavior of the molecules has been correlated to the presence of a specific aliphatic linker including a carbonyl functional group and the effect can be enhanced by the presence of fluorine substituent.
In general the manuscript is well-written, organized and clear. The experimental procedures are systematic and supported by the statistics. Figures are generally clear and appealing even if the graphics of some of them can be improved. However, some points need to be addressed and clarified before considering, from my side, the manuscript suitable for Membranes journal. Below, the list of points need to be addressed
Thank you for the review. As follows you will find a point by point answers.
-The title should be reviewed; the study is mainly experimental so I suggest to remove 'In Silico study' from the title.
The title was edited as required.
-In the introduction authors mention the drug-membrane interaction as a crucial and complex process which may regulate the drug activity; then a detailed and exhaustive description of NSAIDs has been provided. I do think that the importance of the drug-membrane interaction should be even more highlighted by referring to antimicrobial and lipo-peptides (AMPs and LPs), known to interact as a primary step with the lipid membrane. Moreover, their mechanism of action is strongly dependent on both the chemical structure of the peptide and on the lipid composition; these considerations are coherent with the topic of the article and would broaden the discussion beyond the NSAIDs and thus the audience.
-Following the previous point, you should make few consideration on lipopeptide Daptomycin both for its mechanism of action and for its chemical structure which includes Kynurenine an highly fluorescent residue used to localize the peptide-membrane interaction (see following works https://doi.org/10.1021/acs.jpcb.0c06640; https://doi.org/10.1007/s00249-020-01445-w). You should cite this and similar references; tracing the position of drugs within lipid membranes is never a trivial issue and, in this way you would emphasize your work by referring to similar effects with molecules different than NSAIDs.
The text was edited as required. As reviewer’s pointed out, we have added more references. The following paragraph has been added to the introduction:
The interaction of different drugs e.g. antibiotics, nonsteroidal anti-inflammatory drugs (NSAIDs) with lipid membranes is never a trivial issue. Some antibiotics (e.g. daptomycin) and lipopeptides interact as a primary step with the lipid membrane. Their mechanism of action is strongly dependent on both the chemical structure of the peptide and on the lipid composition. Overall, drugs can act on the surface of cell membrane as well as have intracellular targets of action.
-Figure 1 graphics should be improved; the axis size is quite small and labels indicating the kind of derivatives as well as the molar ratio is lacking (the reading is quite tough with such info only in the caption)
-Same consideration are valid for Figure 3.
The Figure 1 and 3 were edited as required.
-In 3.1.3 lines 280-81 authors state: "In both lipids, the transition enthalpy was decreased by PR1, PR12 and PR27 more effectively than by PR2, PR26, PR38 and piroxicam (Figure 5c). " This is not coherent with Figure 5c, which show a different trend depending on the lipids: the decrease seems to be more effective on DMPC. Please clarify
The text in question was rewritten as follows:
In both lipids, the transition enthalpy was decreased by PR1, PR12 and PR27 more effec-tively than by PR2, PR26, PR38 and piroxicam, but the effect was more pronounced for the DMPC (Figure 5c).
- In 3.1.3 lines 281-83 authors state: "Also in both lipids,the main transition temperature was decreased by PR2, PR12 and PR27 more effectively than by PR1, PR38, PR26 and piroxicam (Figure 5a). " Also here the message is not totally coherent with the figure; panel 5a indicates a higher transition temperature decrease with the mentioned derivatives but only on DMPC. Please clarify
The text in question was rewritten as follows:
In both lipids, the main transition temperature was decreased by PR2, PR12 and PR27 more effectively than by PR1, PR38, PR26 and piroxicam, but the effect was also more pronounced for the DMPC (Figure 5a).
- In 3.1.3 lines 283-85 authors state: "Tm was reduced similarly by all studied compounds. The only exceptions were PR2 and PR27, which were the most effective in DMPC bilayers." Again, from the figure it seems that the larger effect is induced by PR2 and PR27 on DMPC lipids but this is not an exception; rather, PR38 seems as an exception as its effect is larger on DPPC. Please clarify
The text in question was rewritten as follows:
Tm was similarly lowered by all studied compounds. The PR2 and PR27 in DMPC bilayers were the most effective ones. The only exception was PR38 which induced larger effect on DPPC.
-lines 333-35 authors state: "The experiments presented revealed that the two–carbon aliphatic linker with a carbonyl group between 4-phenylpiperazine and 1,2-benzothiazine (present in compounds PR2, PR12 and PR27) increased the ability of the compounds to interact deeper" You may add a speculative sentence about the possible role of the charge due to the presence of carbonyl group.
The following sentence has been added to the text: This may suggest an interaction between the lone pair of electrons of the carbonyl group and the model membrane.
-line 369: where does 586KDa come from? Is it MW of PRX - is not present in the tabel. Please clarify
This is a mistake and it has been corrected (should be 522 KDa).
-line 371 author state: "As can be seen compounds PR2 and PR12 exhibit optimal values of logP " Also PR27 show optimal value.
We rewrote it as suggested by the reviewer.
-As conclusive remarks, authors should refer as well to the possibility to evaluate effect of such derivatives forms also on phase-separated lipid systems. This can help in further understanding of the mechanism of action; even a brief referring to such an aspect would highlight the significance of the issue
The following sentence has been added to the text: These interactions could be responsible for at least some of the membrane-dependent effects of oxicam analogues, however the understanding of their importance needs further studies (e.g. the evaluation of effect of these derivatives on phase-separated lipid systems).
Minor
-lines 61-62, use the abbreviation NSAIDs; It has been done.
-line 235, may you refer to Figure 1b and 1d instead of Figure2; It has been done.
-line 421; depend instead of dependent. It has been done.
-line 471: "It can in result determine their bioavailability". Please, rephrase.
The sentence was rewritten as follows:In result these properties determine bioavailability of studied compounds.
Reviewer 2 Report
This is a very meaningful and practical research work. Data research is very systematic, and potential applications are also very extensive. Therefore, I would like to recommend this manuscript for publication after minor revision:
1. Although the Introduction part introduces the research background in great detail, there are too many paragraphs, which are too trivial. It is suggested to integrate the introduction into 3-5 paragraphs.
2. It is suggested to add the potential application of this study in the Introduction part, such as Colorectal disease < Colorectal Cancer and Adjacent Normal Mucosa Differ in Apoptotic and Inflammatory Protein Expression, Engineered Regeneration 2 (2022) 279-287.> or other chronic diseases <The role of astaxanthin on chronic diseases. Crystals 2021; 11: 505.>.
3. Line 183- 183, It is generally believed that the number of samples determines the repeatability of the experiment, and the detection times of each sample represent the precision of the instrument, so they show the credibility of different directions. Generally, we think it is appropriate to set the number of samples at more than 3 rather than 2.
4. Figure 2, in the legend, the authors should add the data present format, eg. (mean ± SD, n=?).
5. Figure 4, in the legend, the authors should add the data present format (mean ± SD, n=?).
6. Figure 7, 8 and 9, in the legend, the authors should add the data present format (mean ± SD, n=?).
Author Response
This is a very meaningful and practical research work. Data research is very systematic, and potential applications are also very extensive. Therefore, I would like to recommend this manuscript for publication after minor revision:
- Although the Introduction part introduces the research background in great detail, there are too many paragraphs, which are too trivial. It is suggested to integrate the introduction into 3-5 paragraphs.
Due to the fact that another reviewer approved the introduction and recommended adding other threads, we chose not to change this section to present the current state of research into the interaction of NSAIDs with membranes.
- It is suggested to add the potential application of this study in the Introduction part, such as Colorectal disease < Colorectal Cancer and Adjacent Normal Mucosa Differ in Apoptotic and Inflammatory Protein Expression, Engineered Regeneration 2 (2022) 279-287.> or other chronic diseases <The role of astaxanthin on chronic diseases. Crystals 2021; 11: 505.>.
The text was edited as required. The following paragraph has been added to the introduction:
There is a need for new, safer than currently used NSAIDs, which will be used both as anti-inflammatory agents, but also as potential chemopreventive agents. With long-term therapy, they could prevent the development of cancer, especially colorectal cancer in patients at increased risk of developing cancer.
- Line 183- 183, It is generally believed that the number of samples determines the repeatability of the experiment, and the detection times of each sample represent the precision of the instrument, so they show the credibility of different directions. Generally, we think it is appropriate to set the number of samples at more than 3 rather than 2.
Thank you for this comment. We started our measurements with two samples for DSC and three for spectroscopy. If the DSC measurement results were significantly different, we performed the measurement for the third sample. We performed each of them in several repetitions. The average for DSC measurements was taken from several repetitions due to the fact that the calorimeter we used is not the most modern and we wanted to be sure that the results are repeatable, and the measurement is precise.
- Figure 2, in the legend, the authors should add the data present format, eg. (mean ± SD, n=?). It has been done.
- Figure 4, in the legend, the authors should add the data present format (mean ± SD, n=?). It has been done.
- Figure 7, 8 and 9, in the legend, the authors should add the data present format (mean ± SD, n=?). It has been done.
Reviewer 3 Report
The manuscript by Maniewska et al. 'Oxicam derivatives interaction with the artificial models of biological membranes – calorimetric, fluorescence spectroscopic and in silico study' seems interesting. The overall design looks good and recommends for publication.
Author Response
The manuscript by Maniewska et al. 'Oxicam derivatives interaction with the artificial models of biological membranes – calorimetric, fluorescence spectroscopic and in silico study' seems interesting. The overall design looks good and recommends for publication.
Thank you for the review and positive feedback on our publication.
Round 2
Reviewer 1 Report
Authors addressed all the points raised up in my previous revision round.